# Inhibitory Effects of Thermolysis Transformation Products of Rotenone on Nitric Oxide Production

**DOI:** 10.3390/ijms24076095

**Published:** 2023-03-23

**Authors:** Gyeong Han Jeong, Hanui Lee, Seung Sik Lee, Byung Yeoup Chung, Hyoung-Woo Bai, Tae Hoon Kim

**Affiliations:** 1Research Division for Biotechnology, Advanced Radiation Technology Institute (ARTI), Korea Atomic Energy Research Institute (KAERI), Jeongeup 56212, Republic of Korea; jkh4598@kaeri.re.kr (G.H.J.); hnlee11@kaeri.re.kr (H.L.); sslee@kaeri.re.kr (S.S.L.); bychung@kaeri.re.kr (B.Y.C.); 2Center for Companion Animal New Drug Development, Korea Institute of Toxicology (KIT), Jeongeup 56212, Republic of Korea; 3Radiation Biotechnology and Applied Radioisotope Science, University of Science and Technology (UST), Daejeon 34113, Republic of Korea; 4Department of Food Science and Biotechnology, Daegu University, Gyeongsan 38453, Republic of Korea

**Keywords:** rotenone, biotransformation, thermal decomposition, nitric oxide, *Derris* roots

## Abstract

Rotenone, isolated from *Derris*, *Lonchocarpus*, and *Tephrosia* from the family Fabaceae, has been shown to have a variety of biological properties and is used in various agricultural industries as a potent biopesticide. However, recent reports have demonstrated that rotenone has the potential to cause several adverse effects such as a neurodegenerative disease. This study aimed to induce thermolysis of the biopesticide rotenone and enhance the functionality of the degraded products. Rotenone (**1**) was degraded after autoclaving for 12 h, and the thermolytic reactants showed enhanced anti-inflammatory capacity against nitric oxide (NO) production. The structures of the newly modified products were spectroscopically determined. The thermal reaction products included various isoflavonoid derivatives **2**–**6**, whose structures were characterized as being produced via chemical reactions in rotenone at the C-12 positions. Among the degraded products, (−)-tubaic acid (**6**) exhibited significantly improved anti-inflammatory effects compared to the original rotenone. Quantitative LC-MS analysis of the major thermolysis products generated in *Derris* extract containing rotenone was performed using isolate **2**–**5** purified from autoclaved rotenone. These results suggest that the thermal transformation of rotenone can improve the functionality of anti-inflammatory agents.

## 1. Introduction

The inflammatory response is a mechanism that defends the biological tissues of human bodies. The normal inflammatory response has a control process in which the production of pro-inflammatory mediators decreases over time, and anti-inflammatory mediators increase, limiting the inflammatory response by itself [1]. Nitric oxide, a product of inflammatory reactions, is produced by NO synthase (NOS) and is an important molecule in immunity and inflammatory reactions [2]. Inducible NOS (iNOS) is expressed in macrophages in an inflammatory environment, and iNOS is not expressed in the normal state. However, when it is expressed by lipopolysaccharides (LPS), a large amount of NO is generated, or inflammatory expression intermediates such as prostaglandin E_2_ (PGE_2_) and tumor necrosis factor-α (TNF-α) are generated [3]. As a result, chronic inflammation and various diseases occur due to NO overexpression by iNOS. Thus, suppression of iNOS and NO production is crucial for the treatment of inflammatory diseases [4]. Therefore, the search for natural compounds that suppress the production or expression of inflammatory mediators is drawing attention as a way to develop anti-inflammatory treatments with fewer side effects [5].

Rotenone, first isolated and identified from *Derris* roots (*Derris elliptica*), is an odorless, colorless isoflavone used as a wide-range insecticide, piscicide, and pesticide [6]. Several isoflavone derivatives recently derived from rotenone were found to exhibit a range of biological activities, including anti-fungal, anti-bacterial, anti-malarial, and anti-cancer properties [7,8]. Park et al. (2012) used the gamma-irradiation technique to induce molecular modification of rotenone, to structurally determined novel rotenoisins A and B [9] and reported potent anti-adipogenic and anti-cancer capacities [10,11,12]. In addition, hydrated rotenone derivatives induced by microbial transformation have been reported to exhibit potent anti-tumor efficacy [13]. As previously reported, superheated steam in the temperature range of 90–190 °C caused rotenone degradation, transforming rotenone into newly degraded products [14]. However, there has been no research related to the isolation, structural identification, and functional evaluation of newly transformed products from thermolysis rotenone.

The present study investigated the transformation processes during the heating of an aqueous solution, under high pressure using an autoclave. The newly modified derivatives of rotenone showed enhanced anti-inflammatory effects against NO production compared with the original rotenone. In addition, we explored the underlying mechanisms and identified a thermolysis reaction for rotenone in a complex extract of *Derris* roots.

## 2. Results and Discussion

### 2.1. Characterization of Newly Formed Products

Rotenone is an odorless toxic isoflavonoid used as an ingredient in natural insecticides and pesticides. Pure rotenone (**1**) was dissolved in an aqueous solution and directly autoclaved for 1, 2, 3, 6, and 12 h. The structural modifications were investigated by reversed-phase high-performance liquid chromatography (HPLC). The inhibitory activity of NO production on LPS-induced RAW 264.7 macrophage cells of the thermally treated reactants increased in a time-dependent manner after exposure to autoclaving for 12 h, in comparison with the original rotenone (Figure 1). Successive column chromatography isolation of the 12 h treated rotenone reactant led to the isolation of the five newly formed products **2**–**6** (Figure 2). The purified compounds were characterized as (−)-rotenolone (**2**), (+)-epirotenolone (**3**), (+)-epirotenone (**4**), dehydrorotenone (**5**), and (−)-tubaic acid (**6**), by a comparison of their spectroscopic data (nuclear magnetic resonance (NMR), electrospray ionization mass spectrum (ESIMS), and optical rotation) with the reference values (Appendix A).

(−)-Rotenolone (**2**): yellow amorphous powder, [α]^20^_D_ −172.7° (*c* 0.1, CHCl_3_), ^1^H NMR (acetone-*d*_6_, 600 MHz): *δ* 7.77 (1H, d, *J* = 8.4 Hz, H-11), 6.64 (1H, s, H-1), 6.54 (1H, d, *J* = 8.4 Hz, H-10), 6.49 (1H, s, H-4), 5.34 (1H, dd, *J* = 9.0, 8.4 Hz, H-2′), 5.07 (1H, br s, H-5′α), 4.92 (1H, br s, H-5′β), 4.70 (1H, d, *J* = 1.2 Hz, H-6a), 4.59 (1H, dd, *J* = 12.0, 1.2 Hz, H-6β), 4.47 (1H, d, *J* = 12.0 Hz, H-6α), 3.77 (3H, s, OCH_3_-2), 3.61 (3H, s, OCH_3_-3), 3.29 (1H, dd, *J* = 15.6, 9.0 Hz, H-3′α), 2.93 (1H, dd, *J* = 15.6, 8.4 Hz, H-3′β), 1.76 (3H, s, H-6′), ^13^C NMR (acetone-*d*_6_, 150 MHz): *δ* 190.6 (C-12), 167.3 (C-9), 157.4 (C-7a), 151.6 (C-3), 148.9 (C-4a), 143.8 (C-2), 143.6 (C-4′), 129.5 (C-11), 113.0 (C-5′), 112.7 (C-8), 111.6 (C-11a), 111.5 (C-1a), 109.2 (C-1), 104.6 (C-10), 101.1 (C-4), 87.6 (C-2′), 76.2 (C-6a), 67.7 (C-12a), 63.7 (C-6), 56.0 (OCH_3_-2), 55.1 (OCH_3_-3), 30.8 (C-3′), 16.3 (C-6′), ESIMS *m/z* 411 [M + H]^+^ (Figure 3) [15].

(+)-Epirotenolone (**3**): yellow amorphous powder, [α]^20^_D_ −444.8° (*c* 0.1, CHCl_3_), ^1^H NMR (acetone-*d*_6_, 600 MHz): *δ* 7.77 (1H, d, *J* = 8.4 Hz, H-11), 6.66 (1H, s, H-1), 6.55 (1H, d, *J* = 8.4 Hz, H-10), 6.48 (1H, s, H-4), 5.37 (1H, dd, *J* = 9.0, 8.4 Hz, H-2′), 5.03 (1H, br s, H-5′α), 4.88 (1H, br s, H-5′β), 4.71 (1H, d, *J* = 1.2 Hz, H-6a), 4.60 (1H, dd, *J* = 12.0, 1.2 Hz, H-6β), 4.47 (1H, d, *J* = 12.0 Hz, H-6α), 3.75 (3H, s, OCH_3_-2), 3.60 (3H, s, OCH_3_-3), 3.35 (1H, dd, *J* = 15.6, 9.0 Hz, H-3′α), 2.86 (1H, dd, *J* = 15.6, 8.4 Hz, H-3′β), 1.72 (3H, s, H-6′), ^13^C NMR (acetone-*d*_6_, 150 MHz): *δ* 190.6 (C-12), 167.3 (C-9), 157.5 (C-7a), 151.6 (C-3), 148.9 (C-4a), 143.8 (C-2), 143.6 (C-4′), 129.5 (C-11), 112.9 (C-5′), 112.8 (C-8), 111.6 (C-11a), 111.4 (C-1a), 109.1 (C-1), 104.6 (C-10), 101.1 (C-4), 87.6 (C-2′), 76.4 (C-6a), 67.7 (C-12a), 63.7 (C-6), 55.9 (OCH_3_-2), 55.1 (OCH_3_-3), 30.7 (C-3′), 16.4 (C-6′), ESIMS *m/z* 411 [M + H]^+^ (Figure 2) [16].

(+)-Epirotenone (**4**): white amorphous powder, [α]^20^_D_ −59.9° (*c* 0.1, CHCl_3_), ^1^H NMR (acetone-*d*_6_, 600 MHz): *δ* 7.78 (1H, d, *J* = 8.4 Hz, H-11), 6.75 (1H, s, H-1), 6.53 (1H, d, *J* = 8.4 Hz, H-10), 6.45 (1H, s, H-4), 5.38 (1H, dd, *J* = 9.0, 7.8 Hz, H-2′), 5.13 (1H, m, H-6a), 5.04 (1H, br s, H-5′α), 4.88 (1H, br s, H-5′β), 4.60 (1H, dd, *J* = 12.6, 3.0 Hz, H-6β), 4.29 (1H, d, *J* = 12.6 Hz, H-6α), 3.91 (1H, d, *J* = 3.6 Hz, H-12a), 3.74 (3H, s, OCH_3_-2), 3.64 (3H, s, OCH_3_-3), 3.38 (1H, dd, *J* = 15.6, 9.0 Hz, H-3′α), 2.92 (1H, dd, *J* = 15.6, 7.8 Hz, H-3′β), 1.75 (3H, s, H-6′), ^13^C NMR (acetone-*d*_6_, 150 MHz): *δ* 188.4 (C-12), 167.0 (C-9), 158.1 (C-7a), 150.2 (C-3), 148.1 (C-4a), 143.9 (C-2), 143.6 (C-4′), 129.4 (C-11), 113.7 (C-5′), 112.9 (C-8), 111.6 (C-11a), 111.4 (C-1a), 105.3 (C-1), 104.2 (C-10), 101.3 (C-4), 87.6 (C-2′), 72.5 (C-6a), 66.1 (C-6), 55.9 (OCH_3_-2), 55.1 (OCH_3_-3), 44.2 (C-12a), 30.8 (C-3′), 16.3 (C-6′), ESIMS *m/z* 395 [M + H]^+^ (Figure 3) [17].

Dehydrorotenone (**5**): yellow amorphous powder, [α]^20^_D_ −48.6° (*c* 0.1, CHCl_3_), ^1^H NMR (acetone-*d*_6_, 600 MHz): *δ* 8.45 (1H, s, H-1), 8.07 (1H, d, *J* = 8.4 Hz, H-11), 6.97 (1H, d, *J* = 8.4 Hz, H-10), 6.61 (1H, s, H-4), 5.53 (1H, dd, *J* = 9.0, 8.4 Hz, H-2′), 5.15 (1H, br s, H-5′α), 5.07 (1H, s, H-6), 4.98 (1H, br s, H-5′β), 3.83 (3H, s, OCH_3_-2), 3.78 (3H, s, OCH_3_-3), 3.65 (1H, dd, *J* = 15.0, 9.0 Hz, H-3′α), 3.25 (1H, dd, *J* = 15.0, 8.4 Hz, H-3′β), 1.62 (3H, s, H-6′), ^13^C NMR (acetone-*d*_6_, 150 MHz): *δ* 173.4 (C-12), 164.8 (C-9), 156.7 (C-7a), 152.4 (C-6a), 149.7 (C-3), 146.7 (C-2), 144.2 (C-4a), 143.6 (C-4′), 127.4 (C-11), 118.4 (C-1a), 113.5 (C-12a), 111.8 (C-1), 111.3 (C-5′), 111.1 (C-11a), 110.6 (C-8), 108.3 (C-10), 100.8 (C-4), 87.7 (C-2′), 64.5 (C-6), 55.8 (OCH_3_-2), 55.2 (OCH_3_-3), 31.7 (C-3′), 16.2 (C-6′), ESIMS *m/z* 393 [M + H]^+^ (Figure 3) [18].

(−)-Tubaic acid (**6**): yellow amorphous powder, [α]^20^_D_ −60.3° (*c* 0.1, CHCl_3_), ^1^H NMR (acetone-*d*_6_, 600 MHz): *δ* 7.74 (1H, d, *J* = 8.4 Hz, H-3), 6.09 (1H, d, *J* = 8.4 Hz, H-4), 5.21 (1H, br t, *J* = 9.0 Hz, H-2′), 5.05 (1H, br s, H-5′α), 4.86 (1H, br s, H-5′β), 3.23 (1H, dd, *J* = 15.0, 9.0 Hz, H-3′α), 2.86 (1H, dd, *J* = 15.0, 8.4 Hz, H-3′β), 1.74 (3H, s, H-6′), ^13^C NMR (acetone-*d*_6_, 150 MHz): *δ* 175.4 (COOH-2), 163.5 (C-5), 159.5 (C-1), 144.8 (C-4′), 131.8 (C-3), 113.0 (C-6), 110.7 (C-2), 110.6 (C-5′), 98.5 (C-4), 86.3 (C-2′), 31.5 (C-3′), 16.4 (C-6′), ESIMS *m/z* 221 [M + H]^+^ (Figure 3) [19].

Isoflavonoid skeleton compounds **2**–**5** were isolated and characterized from *Derris*, *Tephrosia*, and *Lonchocarpus* from the family Fabaceae containing mother rotenone [20,21]. The phenolic acid, (−)-tubaic acid (**6**), was first identified in *Derris* roots [19], and it was reported as a transformed product induced by the alkaline degradation of rotenone [22]. Previous studies have demonstrated that heat treatment is affected by structural modifications that induce chemical reactions such as degradation, hydroxylation, dehydrogenation, and epimerization of flavonoids [23,24]. Our results suggest that thermally induced chemical reactions might be produced under high-pressure steam conditions, and are capable of attacking the chemical bonds at the C-12 position of rotenone, resulting in the generation of rotenone derivatives such as (−)-rotenolone (**2**), (+)-epirotenolone (**3**), (+)-epirotenone (**4**), dehydrorotenone (**5**), and (−)-tubaic acid (**6**) (Figure 3).

### 2.2. Inhibitory Effects of Nitric Oxide Production

The inflammatory reaction is a mechanism by which immune cells recognize external physicochemical stimulation and bacterial infection and secrete various inflammatory mediators to repair and regenerate damaged tissues [25]. Inflammation is closely associated with diverse metabolic syndromes, including neurodegenerative, allergic, degenerative, and cardiovascular diseases [26]. Excessive NO production increases reactive oxygen species (ROS) such as peroxynitrite (ONOO^−^), superoxide anion (O_2_^•−^), and hydrogen peroxide (H_2_O_2_) in the body by generated oxidative stress, which cause inflammation [27]. Recently, one of the most valuable strategies for the therapy of inflammation involves the inhibition of NO production by the disturbing radical formation [3]. Consequently, control of NO production by suppression is considered an important strategy for the development of novel anti-inflammatory materials. The anti-inflammatory effects of natural products, including curcumin, naringenin, and quercetin from natural food stuffs [28], can be estimated using RAW 264.7, based on the inhibitory effects of NO production in in vitro screening (Figure 4).

All isolated pure compounds obtained from thermally treated rotenone were measured for their ability to inhibit NO production in RAW 264.7 macrophage cells for concentrations ranging from 0.625 to 5 μM [29]. No cytotoxicity was observed for any of the tested products at concentrations of up to 5 μM, as measured by the MTT assay (Figure 4A).

Thermolysis of rotenone for 12 h showed enhanced inhibition of NO production on LPS-induced RAW 264.7 cells at a concentration of 5 μg/mL in comparison with the parent rotenone (Figure 1B). As shown in Figure 4B, in untreated RAW 264.7 cells, NO production was almost undetectable. Upon treatment with LPS, the nitrite concentrations in the medium increased markedly. The degradation of compound **6**, at concentrations of 2.5 and 5 μM, significantly inhibited NO production by 59 and 63 μM, respectively, compared to the LPS-induced control group. Hydroxylated rotenone derivatives **2** and **3** exhibited greater potency in inhibiting NO production than the original rotenone. In contrast, (+)-epirotenone (**4**) and hydrorotenone (**5**) showed relatively less potency than compounds **2**, **3**, and **6** produced by the hydroxylation and degradation of rotenone (Figure 4B). These newly formed products might contribute to the improvement in the NO production inhibitory effects of thermally treated rotenone for 12 h. Thus, changes in the anti-inflammatory capacities of thermally treated rotenone might be due to heat processing as a result of the degradation and oxidation of standard rotenone.

### 2.3. Comparative HPLC Analysis of the Major Products Formed from Thermolysis Rotenone

The contents of the isolated compounds in the thermally treated rotenone at 1, 2, 3, 6, and 12 h were quantified using the external standard method, and the results are shown in Table 1. Five concentration points (*n* = 3) were used to prepare the calibration curve, and the calibration curve of the pure solution of the standard compounds was linear (*R^2^* > 0.999). The retention times of the newly transformed (−)-rotenolone (**2**, *t*_R_ 20.1 min), (+)-epirotenolone (**3**, *t*_R_ 19.8 min), (+)-epirotenone (**4**, *t*_R_ 21.6 min), dehydrorotenone (**5**, *t*_R_ 24.2 min), (−)-tubaic acid (**6**, *t*_R_ 18.5 min), and rotenone (**1**, *t*_R_ 21.8 min) were detected for the five thermally treated reactants (Figure 2). Quantitative analysis revealed that the contents of the most potent (−)-tubaic acid (**6**) in the thermolytic rotenone for 12 h was 40.3 ± 0.5 mg/g, which is in accordance with the improved anti-inflammatory effects of the 12 h treated sample (Figure 4). As the heat treatment time of rotenone increased, the production of anti-inflammatory (−)-rotenolone (**2**), (+)-epirotenolone (**3**), and dehydrorotenone (**5**) increased in the thermally treated samples, reaching maximum values of 108.9 ± 0.8, 105.4 ± 0.9, and 258.8 ± 1.5 mg/g, respectively, after 12 h of treatment (Table 1). During the last decade, the structural transformation of rotenone to improve its biological capacity has been achieved using microbial and gamma irradiation [9,13]. Previous studies have revealed that rotenone derivatives induced by hydroxylation and oxidation reactions have potent anti-adipogenic and anti-cancer effects [10,11,12]. In this study, the structural modification of rotenone using thermolysis showed potentially enhanced anti-inflammatory effects against nitric oxide production, indicating that degradation and hydroxylation at the C-12 of rotenone are correlated with an increase in biological capacities.

### 2.4. Comparative HPLC Analysis of the Major Products Formed from Thermolysis Rotenone

An HPLC-DAD-MS analysis method, a proven effective tool for the identification of major natural products [30,31], was developed to achieve good separation and detection of the selected newly generated products by thermolysis *Derris* extracts (Figure 5). After analyzing the *Derris* roots, the major component, rotenone (**1**, *t*_R_ 21.8 min), was detected (Figure 5A). By comparing the retention times, UV profile, and [M + H]^+^ and [M − H]^−^ molecular ions of the peaks, the five transformed compounds were unambiguously identified in thermally treated *Derris* extracts for 12 h. The newly generated rotenone derivatives, (−)-rotenolone (**2**, *t*_R_ 20.1 min), (+)-epirotenolone (**3**, *t*_R_ 19.8 min), (+)-epirotenone (**4**, *t*_R_ 21.6 min), and dehydrorotenone (**5**, *t*_R_ 24.2 min), were detected at 280 nm in the thermally treated *Derris* extracts (Figure 5B and Table 2). The representative pesticide component, rotenone, is easily degraded by high temperature and pressure even in a complex extract state and structurally modified into compounds with enhanced functions. These findings can be useful for the pharmaceuticals industry, as they provide insight into thermal process parameters and the production of structurally modified compounds with enhanced functions. This, in turn, can lead to increased safety and quality as well as increased biological capacities.

## 3. Materials and Methods

### 3.1. Chemicals and Instruments

Rotenone, acetonitrile, methanol, formic acid (HPLC grade), deuterated acetone, deuterium oxide, lipopolysaccharides (LPS), and Griess reagent were purchased from Sigma-Aldrich (St. Louis, MO, USA) and Merck (Darmstadt, Germany). All other reagents and chemicals used in this study were of analytical grade.

Spectra of ^1^H and ^13^C nuclear magnetic resonance (NMR) were measured using a Varian VNS600 instrument (Varian, Palo Alto, CA, USA) operated at 600 and 150 MHz, respectively. Chemical shifts are given in *δ* (ppm) relative to those of acetone-*d*_6_ (*δ*_H_ 2.04; *δ*_C_ 29.8) on a tetramethylsilane (TMS) scale. Mass spectra were measured using Agilent HPLC-MS (Agilent Technologies, Palo Alto, CA, USA). Optical rotation was recorded using a P-2000 polarimeter (JASCO, Tokyo, Japan).

YMC gel ODS AQ 120-50S (particle size 50 μm; YMC Co., Kyoto, Japan) and Sephadex LH-20 (particle size 25–100 μm; GE Healthcare Biosciences AB, Uppsala, Sweden) were used for column chromatography, and a microplate reader (Infinite F200; Tecan Austria GmBH, Grodig, Austria) was used to measure the absorbance. A semi-preparative high-performance liquid chromatography (HPLC) system, the Shimadzu LC-20AD system (Shimadzu, Tokyo, Japan), equipped with a photodiode array detector (PDA, SPD-M20A, Shimadzu, Tokyo, Japan) and a YMC-Pack ODS A-302 column (4.6 mm i.d. × 150 mm, particle size 5 μm; YMC Co., Kyoto, Japan) were used to purify the compounds. LC-MS was performed on an Agilent 6120 mass spectrometer (Agilent Technologies, Palo Alto, CA, USA).

### 3.2. Preparation of Thermolysis Samples

Rotenone (10 mg) in aqueous solution (20 mL) was placed in a glass test bottle and autoclaved at 121 °C for 1, 2, 3, 6, and 12 h, and the newly formed products were immediately analyzed by HPLC. Among the dried reactants, sample solutions containing rotenone with thermal treatment at 121 °C for 12 h showed the most enhanced production inhibitory effects at 57 μM in nitric oxide against RAW 264.7 macrophage cells at a concentration of 5 μg/mL compared to that of original rotenone (Figure 1).

### 3.3. Isolation of Degraded Compounds

Pure rotenone (1 g) was dissolved in 2 L of distilled water and directly treated for 12 h. After cooling, the dissolved solvent was removed, and the resulting residue was purified to isolate the components. A portion (775 mg) was then subjected to column chromatography over a YMC gel ODS-AQ-HG (1 cm i.d. × 36 cm) column with aqueous MeOH, yielding pure compound **6** (33.2 mg, *t*_R_ = 18.5 min) and fractions RT1–RT3. Subfraction RT1 (86.6 mg), which was eluted with 80% MeOH in water, was further purified by preparative HPLC (YMC-prep column, 20 mm i.d. × 250 mm; solvent, 55:45 = H_2_O: MeCN; flow rate: 5 mL/min) to yield compounds **2** (12.8 mg, *t*_R_ 20.1 min) and **3** (13.0 mg, *t*_R_ 19.8 min). Fraction RT2 was separated using a preparative HPLC (YMC-prep column, 20 mm i.d. × 250 mm; solvent, 52:48 = H_2_O: MeCN; flow rate: 5 mL/min) to produce compound **4** (4.7 mg, *t*_R_: 21.6 min). Finally, subfraction RT3 (126.8 mg) was purified using a Sephadex LH-20 gel (1 cm i.d. × 39 cm) column with pure MeOH to obtain compound **5** (73.4 mg, *t*_R_ 24.2 min) (Appendix A). HPLC was performed on a YMC-Pack ODS A-302 column using an elution gradient of 20–100% MeCN in 0.1% HCOOH (detection: 280 nm; flow rate: 1.0 mL/min; oven temperature: 40 °C) (Appendix A).

### 3.4. Cell Culture

The RAW 264.7 cell lines (mouse and macrophage) were obtained from the Korean Cell Line Bank (Seoul National University, Seoul, Republic of Korea). Cells were cultured under sterile conditions at 37 °C in a humidified atmosphere containing 5% CO_2_. The cell culture medium consisted of Dulbecco’s modified Eagle’s medium (DMEM, GIBCO Invitrogen Corp., Carlsbad, CA, USA) supplemented with 10% fetal bovine serum (FBS, GIBCO Invitrogen Corp.) and 1% penicillin/streptomycin (GIBCO Invitrogen Corp.).

### 3.5. Cell Viability Assay

The effects of the newly thermolysis products **2**–**6** (Appendix A) from thermal-treated rotenone on cell viability in RAW 264.7 cells were evaluated by 3-(4,5-dimethylthiazol-2-yl)-2,5-diphenyltetrazolium bromide (MTT) method [32]. The cells were seeded at a density of 5 × 10^4^ cells/well in 96-well plates and incubated for 24 h at 37 °C. The cells were treated with rotenone and the isolated compounds (0.625, 1.25, 2.5, and 5 μM) in dissolved free medium and incubated for 24 h at 37 °C. A solution of MTT (0.5 mg/mL) was added to each well and the plates were incubated for 3 h at 37 °C to allow the reaction to take place. After removal of the culture medium, the formazan produced was dissolved in dimethyl sulfoxide (DMSO), and the absorbance was measured at 570 nm using a spectrophotometer to determine cell viability. The control group was considered to be 100%.

### 3.6. Nitric Oxide Production

The RAW 264.7 cells were plated in a 96-well plate at a density of 5 × 10^4^ cells/well and incubated for 24 h at 37 °C. Cells were pretreated with various concentrations of the isolated compounds for 2 h before incubation with LPS (0.1 μg/mL) for 24 h at 37 °C. Nitric oxide production was determined by the reaction of a macrophage culture supernatant with a Griess reagent (Griess modified). The culture supernatant (100 μL) was mixed with the Griess reagent (100 μL) at room temperature and shaken gently for 20 min. Finally, a microplate reader was used to measure the absorbance of the reactants at 548 nm [30].

### 3.7. Extraction of Derris Root and LC-MS Analysis

Dried roots of *Derris elliptica* were collected in Gyeongsan, Republic of Korea, in September 2016 and identified by Prof. Tae Hoon Kim. An herbarium specimen was deposited at the Natural Products Chemistry Laboratory of Daegu University, Gyeongsan, Republic of Korea. Dried *Derris* roots (10 g) were extracted 3 times with 70% acetone (1 L) at room temperature for 3 days. The crude liquid extract was filtered and concentrated using a rotary evaporator (N-1200A, EYELA, Tokyo, Japan) at 40 °C under a vacuum pressure of 90 mbar. After autoclaving the dried *Derris* extract using the same method, a comparative LC-MS analysis was performed.

HPLC-ESI-MS was performed using an Agilent HPLC system (Agilent Technologies) prepared in binary solvent delivery mode. Chromatographic separation was achieved using an Agilent ZORBAX Eclipse Plus C_18_ column (4.6 mm i.d. × 150 mm; particle size 3.5 μm) (Agilent Technologies) at a column temperature of 40 °C and using a mobile phase containing a mixture of 0.1% HCOOH (in H_2_O) and MeCN. Separation was achieved using a gradient mode of 20% MeCN to 100% MeCN over 30 min at a flow rate of 1.0 mL/min and an injection volume of 5 μL. Mass spectrometry (MS) was performed using an Agilent 6120 mass spectrometer (Agilent Technologies). Electrospray ionization (ESI) mass spectra were acquired in positive and negative ion modes over the *m/z* range of 100–1000. The ESI capillary voltage was set at 1.5 kV, the temperatures of the electrospray source and desolvation gas were 100 °C and 350 °C and the drying gas was at a flow rate of 12 L/min. The ion spray and charging voltages were set to 4000 V and 2000 V, respectively.

### 3.8. Statistical Analysis

Data for in vitro analyses of the inhibitory effects of NO production were analyzed using the Proc GLM procedure of SAS software (version 9.3, SAS Institute Inc., Cary, NC, USA). The results are reported as the least square mean values and standard deviations. Statistical significance was considered at * *p* < 0.05 or ** *p* < 0.01.

## 4. Conclusions

In this study, we investigated the chemical modification of rotenone under high-temperature and high-pressure conditions using chromatographic separation. Our analysis yielded five newly generated product isolates, including two hydroxylated rotenones (**2** and **3**), one epimerized rotenone (**4**), one dehydrogenated rotenone (**5**), and one degraded product (**6**), which have never been isolated or characterized in thermolysis rotenone. These compounds were evaluated for potential anti-inflammatory effects using the suppression of NO production bioassay. Notably, the degraded compound **6** and hydroxylated compounds **2** and **3** showed the most potent suppression of NO production compared to the parent rotenone (**1**). Furthermore, we observed that thermal treatment for 12 h significantly increased the generation of major anti-inflammatory compounds **2**, **3**, and **6** up to 108.9 ± 0.8, 105.4 ± 0.9, and 40.3 ± 0.5 mg/g of absolute content, respectively, while the original rotenone content was remarkably degraded (204.1 ± 1.6 mg/g). These findings suggest that thermolysis can be a viable method for generating new anti-inflammatory compounds from natural sources. Furthermore, our study revealed the successful modification of potential anti-inflammatory isoflavonoids from *Derris* extracts using thermolysis. However, further investigations into the mechanisms of compound formation during heating are warranted to fully understand the potential of thermolysis as a method for generating bioactive compounds.

## Figures and Tables

**Figure 1 ijms-24-06095-f001:**
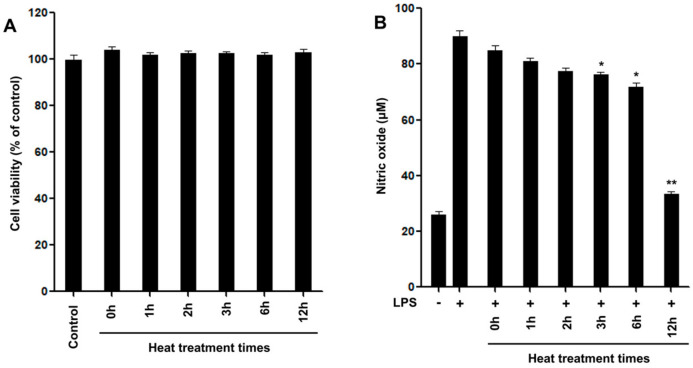
Effects of various thermal treatment reactants (5 μg/mL) of rotenone on LPS-induced RAW 264.7 cells. (**A**) Cell viability detection by MTT assay in RAW 264.7 cells after treatment with heat reaction products for 24 h. (**B**) The cells were treated with samples for 2 h, followed by treatment with the LPS (0.1 μg/mL) for 24 h. The nitrite content of culture media was analyzed. The cells non-treated with reaction products and LPS are referred to as the control group. The experiments were conducted in triplicate and the data are expressed as mean ± SD. * *p* < 0.05, ** *p* < 0.01 as compared to the LPS-induced group.

**Figure 2 ijms-24-06095-f002:**
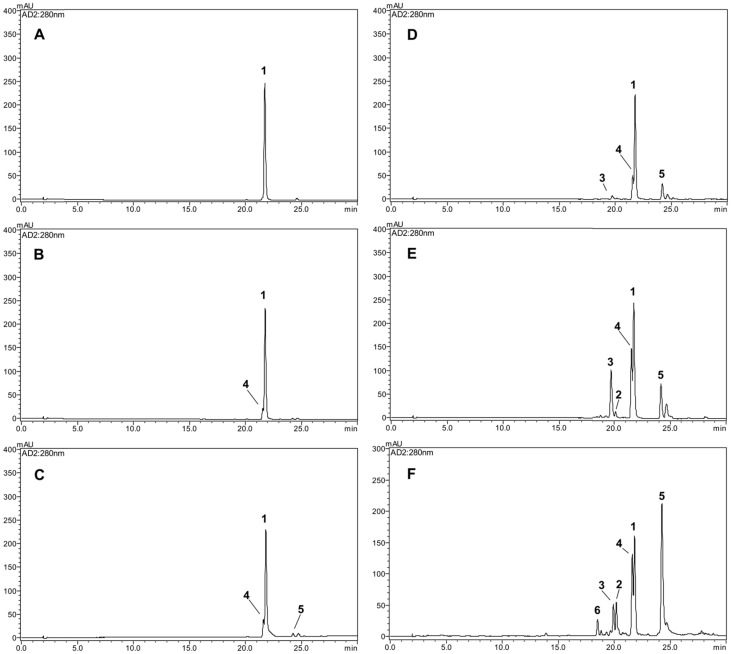
HPLC chromatograms of various thermal reactants (**A**–**F**) of rotenone. (**A**) 0, (**B**) 1, (**C**) 2, (**D**) 3, (**E**) 6, (**F**) 12 h. **1**: rotenone, **2**: (−)-rotenolone, **3**: (+)-epirotenolone, **4**: (+)-epirotenone, **5**: dehydrorotenone, **6**: (−)-tubaic acid.

**Figure 3 ijms-24-06095-f003:**
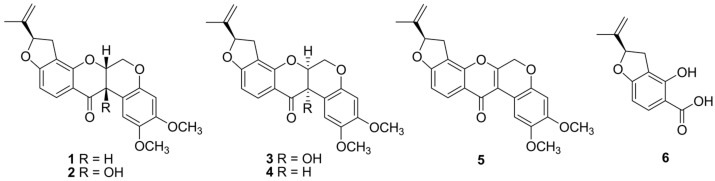
Structures of newly thermolysis products **2**–**6** of rotenone.

**Figure 4 ijms-24-06095-f004:**
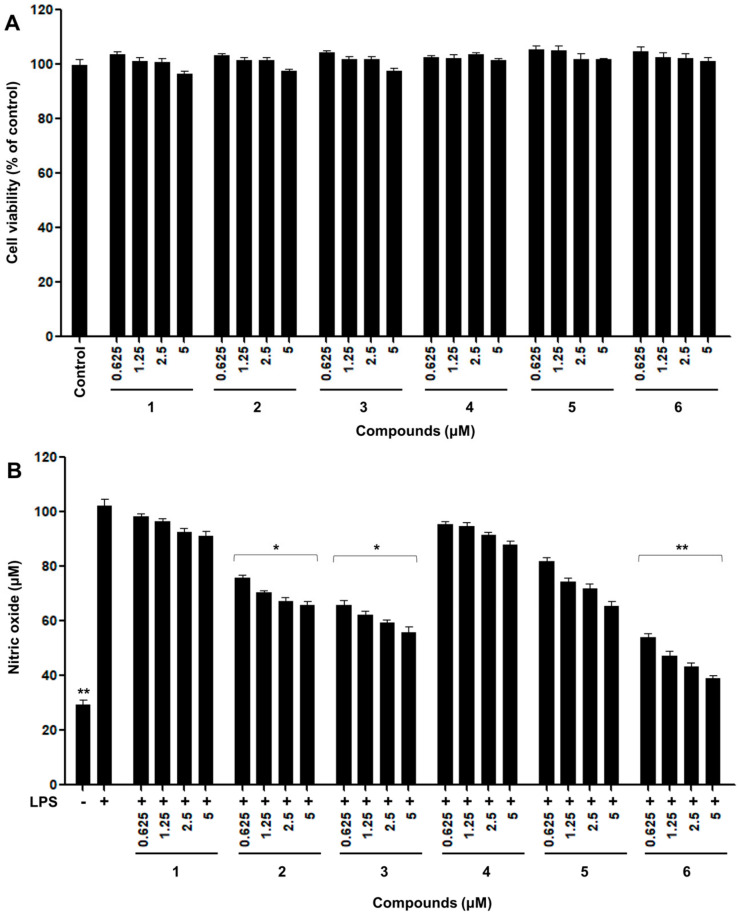
Effects of the isolated compounds **2**–**6** of rotenone on LPS-induced RAW 264.7 cells. (**A**) Cell viability detection by MTT assay in RAW 264.7 cells after treatment with isolated compounds for 24 h. (**B**) The cells were treated with compounds for 2 h, followed by treatment with the LPS (0.1 μg/mL) for 24 h. The nitrite content of culture media was analyzed. The cells non-treated with isolated compounds and LPS are referred to as the control group. The experiments were conducted in triplicate and the data are expressed as mean ± SD. * *p* < 0.05 or ** *p* < 0.01 as compared to the LPS-induced group. **1**: rotenone, **2**: (−)-rotenolone, **3**: (+)-epirotenolone, **4**: (+)-epirotenone, **5**: dehydrorotenone, **6**: (−)-tubaic acid.

**Figure 5 ijms-24-06095-f005:**
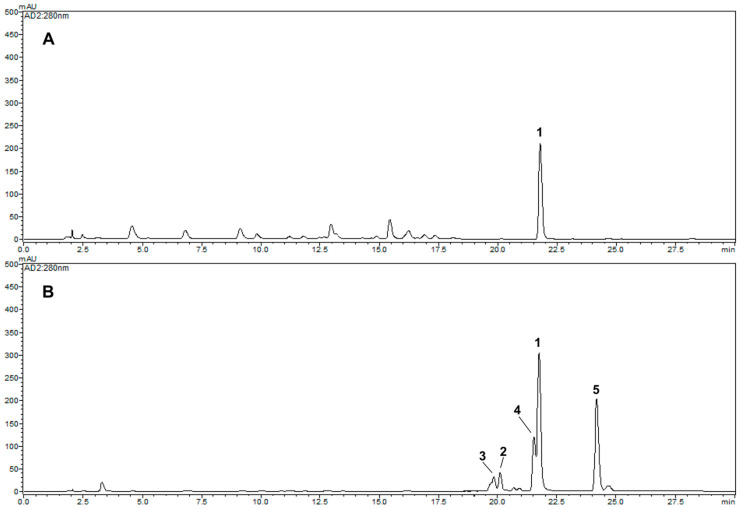
HPLC chromatograms of thermal reactant (**B**) of the *Derris* root extract (**A**). (**A**) *Derris* root extract, (**B**) heat-treated reactant for 12 h. **1**: rotenone, **2**: (−)-rotenolone, **3**: (+)-epirotenolone, **4**: (+)-epirotenone, **5**: dehydrorotenone.

**Table 1 ijms-24-06095-t001:** Content (mg/g) of individual components in the reaction mixture by thermal treatment times.

Compounds	*t*_R_ (min)	Absolute Content (mg/g) ^1^				
		1 h Reactant	2 h Reactant	3 h Reactant	6 h Reactant	12 h Reactant
Rotenone (**1**)	21.8	952.3 ± 3.2 ^a^	892.6 ± 2.9 ^a^	805.3 ± 2.7 ^a^	477.6 ± 1.8 ^ab^	204.1 ± 1.6 ^b^
**2**	20.1	ND ^2^	ND	ND	13.5 ± 0.3 ^f^	108.9 ± 0.8 ^c^
**3**	19.8	ND	ND	10.7 ± 0.3 ^f^	99.7 ± 0.8 ^c^	105.4 ± 0.9 ^c^
**4**	21.6	12.3 ± 0.2 ^f^	30.1 ± 0.2 ^e^	42.3 ± 0.5 ^d^	165.6 ± 1.0 ^bc^	186.9 ± 1.0 ^bc^
**5**	24.2	ND	8.3 ± 0.1 ^f^	39.7 ± 0.5 ^d^	87.0 ± 0.7 ^cd^	258.8 ± 1.5 ^b^
**6**	18.5	ND	ND	ND	ND	40.3 ± 0.5 ^d^

^1^ All the compounds were examined in triplicate. Means with different letters (a–f) within the column differ significantly (*p* < 0.05). ^2^ ND: Not detected.

**Table 2 ijms-24-06095-t002:** LC−MS data of the major bioactive constituents identified in thermal-treated *Derris* root extract.

*t*_R_ (min)	UV λ_max_ (nm)	MF	[M + H]^+^	[M − H]^−^	Identification
19.8	242, 303	C_23_H_22_O_7_	411	409	(+)-Epirotenone (**3**)
20.1	240, 305	C_23_H_22_O_7_	411	409	(−)-Rotenolone (**2**)
21.6	218, 230, 294	C_23_H_22_O_6_	395	393	(+)-Epirotenolone (**4**)
21.8	217, 234, 290	C_23_H_22_O_6_	395	393	Rotenone (**1**)
24.2	228, 280, 310	C_23_H_20_O_6_	393	392	Dehydrorotenone (**5**)

## Data Availability

The data presented in this study are available upon request from the corresponding author.

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
