# Peer review of "Inhibitory Effects of Thermolysis Transformation Products of Rotenone on Nitric Oxide Production"

_ijms, 2023, doi:10.3390/ijms24076095_

Round 1

Reviewer 1 Report

In this manuscript the chemical modification of rotenone
under high-temperature and high-pressure
conditions using chromatographic separation has been presented.
I am sending a list of necessary corrections in
the manuscript that should be made before publication.

Sincerely,

Author Response

Reviewer 1

In this manuscript the chemical modification of rotenone under high-temperature and high-pressure conditions using chromatographic separation has been presented. I am sending a list of necessary corrections in the manuscript that should be made before publication.

Point 1: Line 89. To correct bracket

Response 1: We corrected bracket

Point 2: Figure 2. The labels on the X and Y axes should be clearer

Response 2: We changed high quality figure.

Point 3: Line 172 : You should check the order of the  figures.  The label for image four should be corrected. There are references to Figure 4 several times. Lines: 171, 185, 188,192, 196… 

Response 3: We checked figure numbers and corrected Figure 4.

Point 4: Line 340: Check the Celsius markings

Response 4: We corrected Celsius markings and showed as red-color.

The authors wish to thank the reviewers for the constructive and helpful comments on the revision of this article.

Reviewer 2 Report

In the article, the authors presented investigate the chemical modification of rotenone under high-temperature and high-pressure conditions using chromatographic separation. These studies were conducted to demonstrate that thermolysis could be a viable method for producing new anti-inflammatory compounds from natural sources.

I believe that the studies presented in this work are useful. I do not have any remarks. The article is written in an appropriate way. Therefore, I suggest to accept the paper for publication in International Journal of Molecular Sciences

There are only two typos that need to be corrected:

  1. page 4, line 157 – “prodution” – it should be "production",
  2. page 3, line 69 – “compunds” – it should be "compounds".

Author Response

Reviewer 2

In the article, the authors presented investigate the chemical modification of rotenone under high-temperature and high-pressure conditions using chromatographic separation. These studies were conducted to demonstrate that thermolysis could be a viable method for producing new anti-inflammatory compounds from natural sources.

I believe that the studies presented in this work are useful. I do not have any remarks. The article is written in an appropriate way. Therefore, I suggest to accept the paper for publication in International Journal of Molecular Sciences

There are only two typos that need to be corrected:

Point 1: page 4, line 157 – “prodution” – it should be "production"

Response 1: We corrected word and showed as red-color.

Point 2: page 3, line 69 – “compunds” – it should be "compounds".

Response 2: We corrected word and showed as red-color.

The authors wish to thank the reviewers for the constructive and helpful comments on the revision of this article.
